# Long-term impact of 10-valent pneumococcal conjugate vaccine among children <5 years, Uganda, 2014–2021

**Mercy Wendy Wanyana**[1]*, **Richard Migisha**[1], **Patrick King**[1], **Lilian Bulage**[1], **Benon Kwesiga**[1], **Daniel Kadobera**[1], **Alex Riolexus Ario**[1,2], **Julie R. Harris**[3]

1 Uganda Public Health Fellowship Program, Uganda National Institute of Public Health, Kampala, Uganda, 2 Ministry of Health, Kampala, Uganda, 3 US Centers for Disease Control and Prevention, Kampala, Uganda

* mwanyana@uniph.go.ug

## Abstract

Pneumonia is the second leading cause of hospital admissions and deaths among children <5 years in Uganda. In 2014, Uganda officially rolled out the introduction of the pneumococcal conjugate vaccine (PCV) into routine immunization schedule. However, little is known about the long-term impact of PCV on pneumonia admissions and deaths. In this study, we described the trends and spatial distribution of pneumonia hospital admissions and mortality among children <5 years in Uganda, 2014–2021. We analysed secondary data on pneumonia admissions and deaths from the District Health Information System version 2 during 2014–2021. The proportion of pneumonia cases admitted and case-fatality rates (CFRs) for children <5 years were calculated for children <5 years presenting at the outpatient department. At national, regional, and district levels, pneumonia mortality rates were calculated per 100,000 children <5 years. The Mann-Kendall Test was used to assess trend significance. We found 667,122 pneumonia admissions and 11,692 (2%) deaths during 2014–2021. The overall proportion of pneumonia cases admitted among children <5 years was 22%. The overall CFR was 0.39%, and the overall pneumonia mortality rate among children <5 years was 19 deaths per 100,000. From 2014 to 2021, there were declines in the proportion of pneumonia cases admitted (31% to 15%; p = 0.051), mortality rates (24/100,000 to 14 per 100,000; p = 0.019), and CFR (0.57% to 0.24%; p = 0.019), concomitant with increasing PCV coverage. Kotido District had a persistently high proportion of pneumonia cases that were admitted (>30%) every year while Kasese District had persistently high mortality rates (68–150 deaths per 100,000 children <5 years). Pneumonia admissions, mortality, and case fatality among children <5 years declined during 2013–2021 in Uganda after the introduction of PCV. However, with these trends it is unlikely that Uganda will meet the 2025 GAPPD targets. There is need to review implementation of existing interventions and identify gaps in order to highlight priority actions to further accelerate declines.

**Data availability statement:** The relevant data belong to the Uganda District Health Information System (Ministry of Health, Republic of Uganda), version 2 database, and are available at https://hmis2.health.go.ug/.

**Funding:** The authors received no specific funding for this work.

**Competing interests:** The authors have declared that no competing interests exist.

## Background

Pneumonia, a largely preventable disease, persists as a major public health problem among children <5 years. In 2019, pneumonia was the leading infectious cause of death globally among children <5 years old, accounting for 14% of all deaths in this age group [1]. Half of the pneumonia cases and deaths among children <5 years are reported in sub-Saharan Africa [2]. While pneumonia incidence among children <5 years declined globally during 2000 to 2015, the same time period saw a three-fold increase in pneumonia requiring admissions in sub-Saharan Africa [2,3]. In Uganda, pneumonia is the second leading cause of all hospital admissions among children <5 years [4].

To reduce this burden, in 2013 Uganda and other low-income countries committed to implementing the World Health Organization's integrated Global Action Plan for the Prevention and Control of Pneumonia and Diarrhoea (GAPPD) [5]. This initiative included the introduction of the pneumococcal conjugate vaccine (PCV). This was because most childhood pneumonia is commonly caused by *Streptococcus pneumoniae* globally [6]. PCV contains serotypes that are commonly associated with invasive and mucosal pneumonia among children <5 years and can reduce both incidence and severity of disease leading to reductions in admissions and mortality [7].

In 2014, Uganda officially rolled out to the 10-valent pneumococcal conjugate vaccine, PCV-10 in the national immunisation schedule [8]. This seemed promising as PCV-10 targets the commonest pneumococcal serotypes documented in Uganda before its introduction (6B,19F, and 23F) as well as serotypes 1, 4, 5, 7F,9V,14, and 18C [9,10]. Three doses of PCV-10 (PCV1, 2, and 3) are given to children at 6, 10, and 14 weeks of age [11]. The introduction of PCV-10 was coupled with other low-cost interventions included in GAPPD namely exclusive breastfeeding for the first six months and continued breastfeeding with appropriate complementary feeding thereafter; use of simple, standardized guidelines for the identification and treatment of pneumonia in the community through integrated community case management of childhood illnesses; and reduction of household air pollution with improved stoves [5]. With these interventions, implemented, Uganda aimed to reduce the incidence of pneumonia requiring admission by 75% in children <5 years from 2013 to 2025 and reduce mortality from pneumonia in children <5 years of age to <3 per 1,000 live births by 2025.

Currently, little is documented on Uganda's progress towards these goals. Previous studies conducted in Uganda assessed the prevalence of pneumonia at specific timepoints and may not reflect Uganda's progress over time [12,13]. Additionally, these studies were conducted in sub-regions and may not be generalisable to the entire country or show meaningful spatial differences. We assessed the temporal trends and spatial distribution of pneumonia admissions and mortality among children <5 years in Uganda from 2014–2021 to assess progress towards these goals.

## Methods

### Study setting

We utilized pneumonia data generated from all health facilities in Uganda. The Uganda health system classifies health facilities into various levels based on their capacities. In Uganda, outpatient pneumonia cases are managed at all levels, while those with pneumonia requiring admission are managed at health centres III and IV, general and regional referral hospitals, and national referral hospitals, all of which have in-patient services [14]. Patients are also referred as needed from lower-level facilities (health centres) to higher-level facilities (hospitals).

## Data source

We conducted a descriptive study using routinely-collected pneumonia surveillance data from the District Health Information System version 2 (DHIS2) during January 2014 to December 2021. DHIS2 is an electronic database that was officially adopted in 2012 [15]. It contains data on priority diseases, conditions, and events of public health importance, including pneumonia from 2013 to-date [16]. To ensure data quality within DHIS2,data in physical health registers are routinely compared with summary reports within the electronic system in addition to the in-built quality checks [15]. Aggregate data on pneumonia cases, admissions, and deaths from both outpatient and inpatient monthly reports (Health Management Information System forms [HMIS] 105 and HMIS 108) from 2013–2021 were used for this study.

## Study variables, data management, and analysis

We obtained aggregate data on pneumonia cases and admissions among children <5 years to calculate the annual proportion of pneumonia cases admitted. A pneumonia admission was defined as a hospital stay in a person with pneumonia as a primary diagnosis, based on the International Classification of Disease-10 framework. A pneumonia case at the outpatient department was defined as pneumonia as a primary diagnosis based on the International Classification of Disease-10 framework in a patient not requiring a hospital stay. We calculated the proportion of pneumonia cases admitted using pneumonia admissions <5 years as a numerator and total pneumonia cases <5 years presenting at the outpatient department as a denominator. (All admitted patients with pneumonia pass through the outpatient department before admission.) Pneumonia deaths were defined as in-patient deaths with pneumonia recorded as the primary cause of death. We calculated pneumonia mortality using pneumonia deaths among children <5 years as a numerator and estimated annual population data for children <5 years from the Uganda Bureau of Statistics (UBOS) as a denominator. Case-fatality rates (CFRs) were calculated as the proportion of pneumonia cases <5 years at health facilities who died at the facility. Reporting rates for cases and deaths were calculated as the percentage of the expected monthly reports that were submitted to DHIS2 from 2013 to 2021. PCV3 vaccine coverage was calculated as the percentage of the target population who received 3 doses of the PCV-10 in a given year based on data obtained from DHIS2. Although the PCV-10 vaccine was introduced in 2013, official rollout to the entire country started in 2014 [8]. PCV3 vaccine coverage was therefore calculated from 2014 to 2021.

We downloaded data from the DHIS2 platform as in Excel file and imported it into EpiInfo 7 software (CDC, Atlanta, USA) for analysis. Annual proportion of pneumonia cases admitted, mortality rates, and case-fatality rates were calculated at national and regional levels. Line graphs were used to describe national and regional trends. The Mann-Kendall test was conducted to assess trends in annual proportion of pneumonia cases admitted, mortality rates, and case-fatality rates. The Z value > 0 and $P < 0.05$ indicated an increased trend while the Z value >0 and $P < 0.05$ represented a decreased trend. Choropleth maps drawn using QGIS software were used to show the spatial distribution of pneumonia admissions and mortality in the country.

## Ethical considerations

This study was done to inform public heath practice and therefore determined as non-research and therefore waived of the full institutional review board. The Ministry of Health (MoH) of Uganda through the office of the Director General Health Services gave approval to access data from the national electronic surveillance data base DHIS2. Our study utilized routinely generated aggregated surveillance data with no personal identifiers in health facility in-patient monthly reports through the DHIS2. We stored the abstracted data set in a password-protected

computer and only shared it with the investigation team. This activity was reviewed by CDC and was conducted consistent with applicable federal law and CDC policy. §See, e.g., 45 C.F.R. part 46, 21 C.F.R. part 56; 42 U.S.C. §241(d); 5 U.S.C. §552a; 44 U.S.C. §3501 et seq.

## Results

### National trends in the proportion of pneumonia cases admitted among children <5 years with pneumonia in Uganda, 2014–2021

During the study period, reporting rates for both admissions and deaths increased from 82% to 92% (Kendall's score = 24, p = 0.017). There was a total of 2,535,553 outpatient pneumonia cases and 667,122 admitted pneumonia cases reported among children <5 years. The overall proportion of pneumonia cases <5 years admitted was 22% (range: 15–34%). Overall, there was a 53% decline in the proportion of pneumonia cases admitted over the study period, from 31% to 15% (Kendall's score = −16, p = 0.051), concurrent with increasing PCV3 vaccine coverage from 60% to 91% (Fig 1A).

### National trends in pneumonia mortality among children <5 years, Uganda, 2014–2021

During the study period, there were 11,892 pneumonia-related deaths among children <5 years. The overall pneumonia mortality rate was 19 deaths per 100,000 children <5 years. There was a 42% decline in pneumonia mortality rates from 24 to 14 per 100,000 children <5 years (Kendall's score = −20, p = 0.019) (Fig 1B).

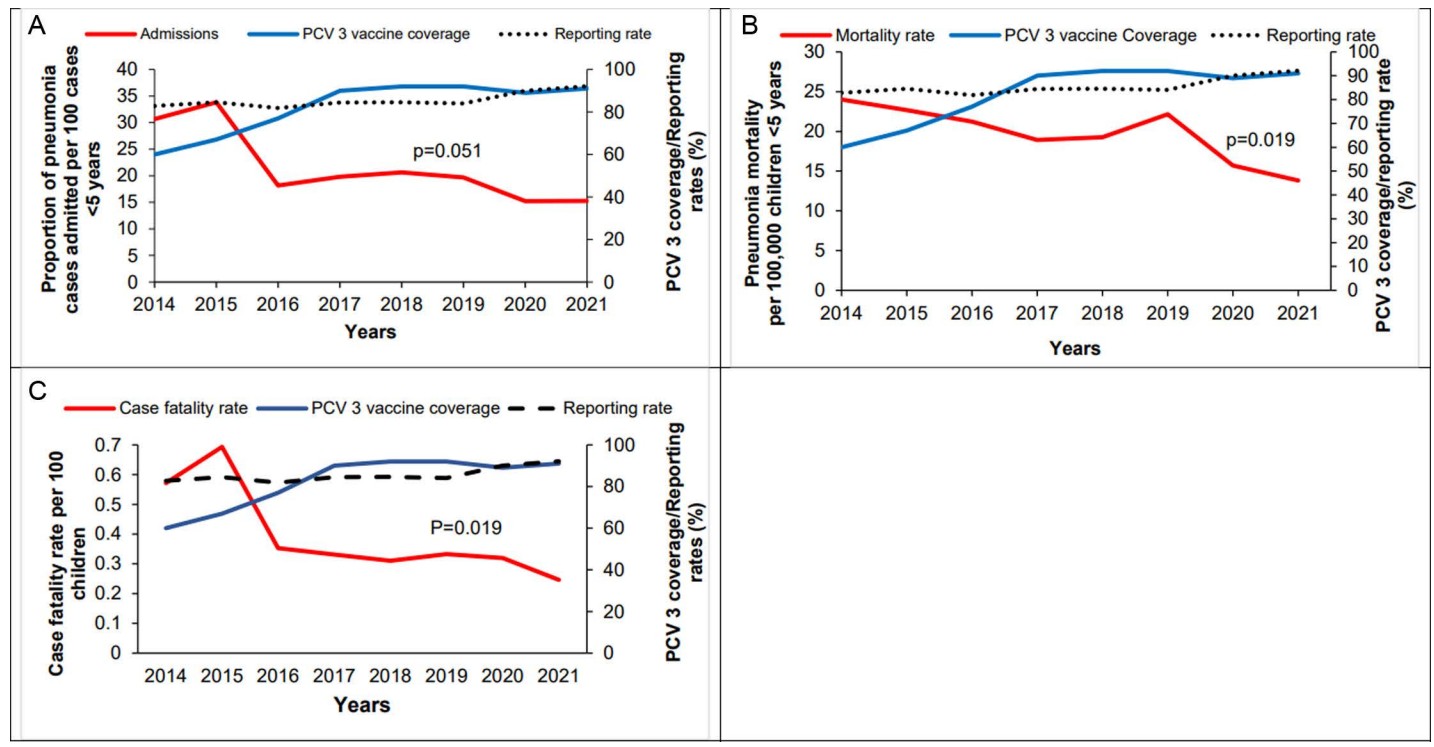

**Fig 1. A: National trend in proportion of pneumonia cases admitted among children <5 years with pneumonia, Uganda, 2014–2021. B: National trend in pneumonia mortality among children <5 years, Uganda, 2014–2024. C: Trends in pneumonia case fatality rate among <5 years, Uganda, 2014–2021; PCV: Pneumococcal conjugate vaccine.**

## National trends pneumonia case-fatality rate among children <5 years, Uganda, 2014–2021

The overall CFR was 0.39% children <5 years with pneumonia (Range: 0.25–0.69). There was a 58% decline from 0.57% to 0.24% over the evaluation period (Kendall's score = −20, p = 0.019) (Fig 1C).

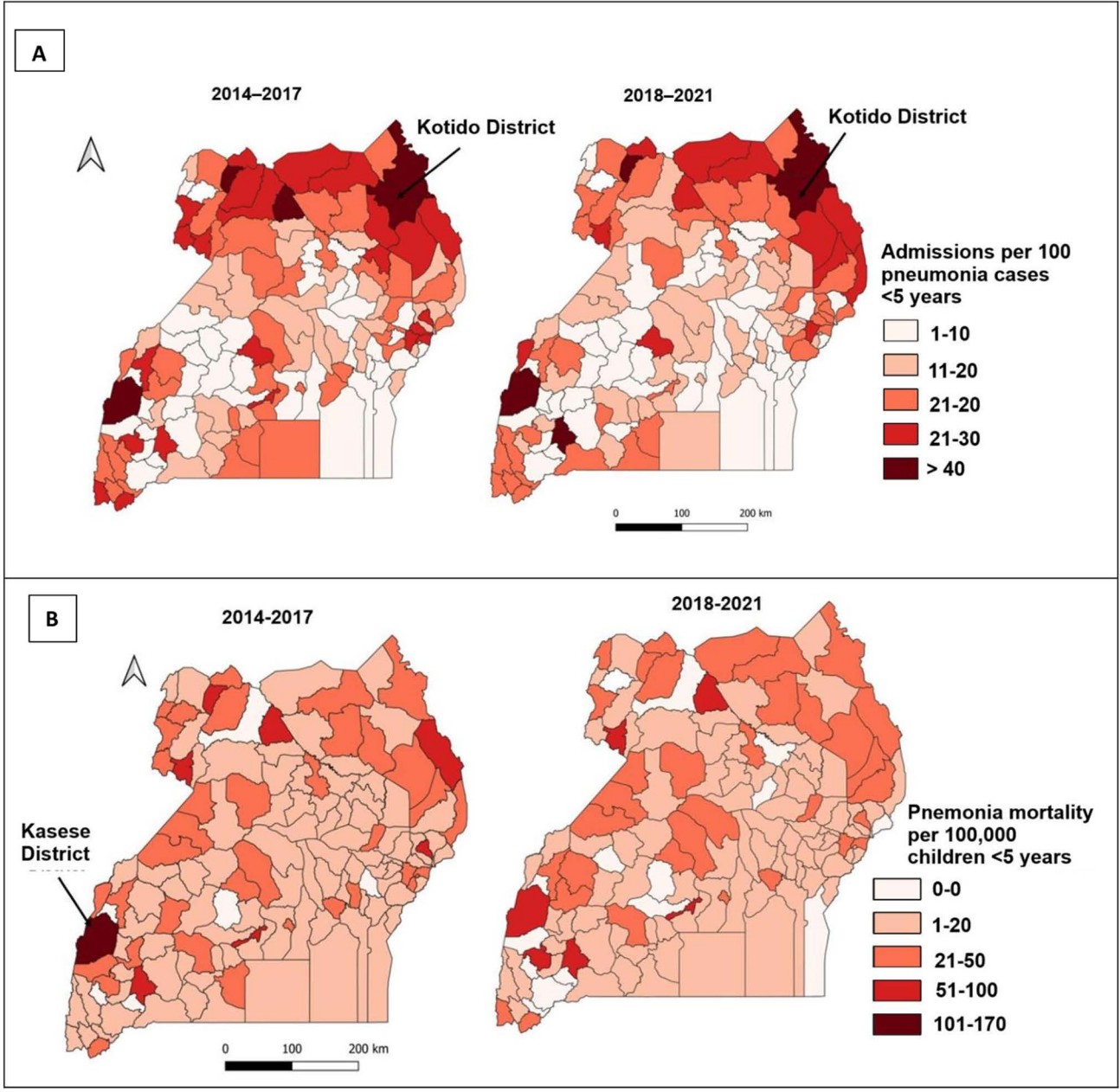

**Fig 2.  A: Spatial distribution of proportion of pneumonia cases admitted among children <5 years, 2014–2021, Uganda. Map derived from QGIS Desktop 3.34.10, 8 September 2022, Shape file source: Uganda Bureau of Statistics, 2021; URL: https://data.unhcr.org/en/documents/details/83043. B: Spatial distribution of pneumonia mortality rate among children <5 years, 2014–2021, Uganda. Map derived from QGIS Desktop 3.34.10, 8 September 2022, Shape file source: Uganda Bureau of Statistics, 2021; URL: https://data.unhcr.org/en/documents/details/83043.**

## Regional trends of pneumonia cases admitted, mortality rate and case fatality rate among children <5 years, Uganda, 2014–2021

Both the proportion of pneumonia cases admitted and pneumonia mortality rates among children <5 years were higher in the Northern Region than in the Western, Eastern, and Central Regions (Table 1). Case-fatality rates were higher in the Central, Northern, and Western regions than in the Eastern Region. Declines in mortality rates were observed over the study period across all regions, while case-fatality rates declined in the Northern and Western regions (Table 1).

## Spatial distribution of proportion of pneumonia cases admitted and mortality among children <5 years, Uganda, 2014–2021

Proportion of pneumonia cases among children <5 years that were admitted was generally highest in north-eastern Uganda. Kotido District had consistently high admissions rates of >30% throughout the study period (Fig 2A).

Kasese District (Fig 2B) had consistently high pneumonia mortality throughout the review period ranging from 68 to 150 deaths per 100,000 children <5 years.

## Discussion

We described the trends and spatial distribution of pneumonia hospital admissions, mortality, and case-fatality among children <5 years in Uganda from 2014–2021. Over the 8 years, there was a decline in both the proportion of pneumonia cases that were admitted and deaths concomitant with an increase in PCV3 coverage.

After the official roll-out of PCV-10 in Uganda in 2014, we found a ≥ 40% reduction in pneumonia cases admitted, mortality, and case-fatality rates among children <5 years over the next 8 years. These findings are in agreement with those in other African countries after PCV was introduced: a study in Burkina Faso identified a 34% reduction in pneumonia admissions among children <5 years five years after the introduction of PCV [17], while another in Zambia found a 38% and 29% decline in pneumonia admissions among children aged <1 year and 1–4 years, respectively, three years after introduction of PCV [18]. Studies in South Africa and Rwanda have had similar findings [19,20]. The greater reductions in our study, compared to the others, could be related to the longer period of observation after the introduction of PCV in our study. Declines in admissions and case fatality rates observed in 2015 could also be attributable to the impact of the WHO-recommended admission criteria introduced in 2014 [21]. Before 2014, any child <5 years old with pneumonia (i.e., with fast breathing and/or chest indrawing) was recommended to be admitted. With the revisions in 2014, recommendations for admission were changed to pneumonia plus any danger signs, which would have led to declines in admissions. The revised treatment guidelines promoted

**Table 1. Regional trends of pneumonia cases admitted, mortality rate, and case fatality rate among children <5 years Uganda, 2014–2021. Kendall's score and p-value represents the presence or absence of an 8-year linear trend.**

| Region | Proportion of pneumonia cases admitted (%) children <5 years | | | Case-fatality rate % | | | Mortality rate per 100,000 children <5 years | | |
|---|---|---|---|---|---|---|---|---|---|
| | Annual Mean | Kendall's Score | P-value | Annual Mean | Kendall's Score | P-value | Annual Mean | Kendall's Score | P-value |
| Central | 14.1 | −4 | 0.70 | 0.49 | −5 | 0.55 | **19** | **−18** | **0.03** |
| Eastern | 10.5 | −11 | 0.71 | 0.33 | −11 | 0.13 | **15** | **−20** | **0.05** |
| Northern | 28.2 | 1 | 0.99 | **0.49** | **−15** | **0.04** | **28** | **−30** | **0.003** |
| Western | 21.4 | −5 | 0.56 | **0.44** | **−15** | **0.04** | **21** | **−26** | **0.009** |

outpatient treatment of pneumonia without danger signs with oral amoxicillin. This was a more effective and accessible treatment, and has been linked to a reduction in pneumonia deaths among children [21].

We noted that the largest decline in case-fatality rates occurred from 2015 to 2016. It is unknown why this occurred, but it may be related to the 32% decline in new HIV infections between 2014 and 2015, which represented the largest decline in many years [22]. HIV is a key risk factor for pneumonia deaths among children <5 years linked and is associated with a 4-fold increased risk of pneumonia death [23]. Additional declines in pneumonia case-fatality rates even after 2015 might also reflect improvements in the quality of healthcare during the study period. Integrated community case management introduced at the start of the review period facilitated prompt diagnosis and access to antibiotics at the community level, improving appropriate care-seeking [24]. During the review period, there were also increases in access to electricity (both hydroelectricity and solar), especially at low-level health facilities over time, there is increased access to oxygen therapy thereby improving patient survival thus reducing case-fatality rates [25,26].

The 50% decline in pneumonia admissions over the 9-year period was borderline significant, suggesting a smaller effect of the PCV on pneumonia admissions than on mortality or case-fatality [27]. While PCV reduces the severity and incidence of pneumonia among the strains it targets [28], previous studies have shown that the introduction of PCV can lead to an increase in the proportion of pneumococcal pneumonia due to non-vaccine pneumococcal serotypes [29,30]. It is possible that in Uganda, like other African countries, other serotypes are taking the place of those targeted by PCV. A study in the Gambia indicated a 47% increase in pneumococcal pneumonia due to non-vaccine serotypes 3 years following PCV13 introduction [31]. Similarly, a 27% increase in pneumococcal pneumonia was observed in Botswana following introduction of PCV [32]. However, there are few data available on the current distribution of circulating serotypes in Uganda. The decline suggests that Uganda may be on track to reach the GAPPD target of a 75% reduction in admissions by 2025. Nonetheless, monitoring of trends in the circulating pneumococcal serotypes in the post-PCV era and maintaining other interventions to reduce the burden and impact of childhood pneumonia is important to ensuring that the reductions continue [33].

Regional variations in the proportion of pneumonia cases admitted, pneumonia mortality, and case-fatality rates were observed. The highest proportion of pneumonia cases admitted and mortality were observed in the northern and eastern regions. These regions are characterised with higher levels of poverty and poor housing compared to other areas of the country, which are known to be associated with increased likelihood of pneumonia requiring admission and mortality [34,35]. There is a need to understand the factors associated with high burden of severe pneumonia in these regions to develop targeted interventions.

Our findings should be interpreted with the following limitations. We used inpatient data, which could lead to underestimation of the true pneumonia mortality by missing cases and deaths that occurred in communities. Secondly, we used aggregated secondary data, which lacked key variables to further explore trends across subcategories within this age group. Thirdly, we were unable to make before-and-after PCV-10 comparisons of pneumonia admissions and deaths due to the lack of data before PCV-10 introduction. As a result, we cannot assess the impact of the introduction of PCV-10 on the observed trends of pneumonia admissions and deaths to the interventions. Finally, despite reductions in pneumonia admissions and deaths with increasing PCV-10 coverage, we cannot definitively attribute these reductions to PCV-10. During this time period, other interventions, including exclusive breastfeeding for the first six months and continued breastfeeding with appropriate complementary feeding thereafter, increased access to antibiotics through integrated community case management of

childhood illnesses, reduction of household air pollution with improved stoves, and increased access to oxygen therapy were implemented, and these also likely led to reductions.

## Conclusion

Our findings demonstrate declines in pneumonia admissions, mortality rates, and case-fatality rates among children <5 years over the 9-year period following the introduction of PCV in Uganda. However, with these trends it is unlikely that Uganda will meet the 2025 targets. Reviewing the implementation of existing interventions and identification of gaps to highlight priority actions could further accelerate decline. We recommend future studies monitoring of trends in the circulating pneumococcal serotypes in the post-PCV.

## Acknowledgments

We would like to thank the Ministry of Health for providing access to DHIS2 data that was used for this analysis. We appreciate the technical support provided by the Division of Surveillance, Information and Knowledge Management, MoH. Finally, we thank the US-CDC for supporting the activities of the Uganda Public Health Fellowship Program.

## Author contributions

**Conceptualization:** Mercy Wendy Wanyana.

**Data curation:** Mercy Wendy Wanyana.

**Formal analysis:** Mercy Wendy Wanyana, Patrick King.

**Methodology:** Mercy Wendy Wanyana, Julie R. Harris.

**Supervision:** Richard Migisha, Benon Kwesiga.

**Visualization:** Mercy Wendy Wanyana.

**Writing – original draft:** Mercy Wendy Wanyana.

**Writing – review & editing:** Mercy Wendy Wanyana, Richard Migisha, Lilian Bulage, Benon Kwesiga, Daniel Kadobera, Alex Riolexus Ario, Julie R. Harris.

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
