## [Decision Letter · Decision Letter 0]

8 Jul 2024

PGPH-D-24-00245

Trends and spatial distribution of pneumonia admissions and deaths among children <5 years, Uganda, 2013–2021

Dear Dr. Wanyana,

Thank you for submitting your manuscript to PLOS Global Public Health. After careful consideration, we feel that it has merit but does not fully meet PLOS Global Public Health’s publication criteria as it currently stands. Therefore, we invite you to submit a revised version of the manuscript that addresses the points raised during the review process.

Please submit your revised manuscript by September 3, 2024. If you need more time than this to complete your revisions, please reply to this message or contact the journal office at globalpubhealth@plos.org. Please include the following items when submitting your revised manuscript:

We look forward to receiving your revised manuscript.

Kind regards,

Dandara de Oliveira Ramos, PhD

Academic Editor

Journal Requirements:

2. Figure 5: please (a) provide a direct link to the base layer of the map (i.e., the country or region border shape) and ensure this is also included in the figure legend; and (b) provide a link to the terms of use / license information for the base layer image or shapefile. We cannot publish proprietary or copyrighted maps (e.g. Google Maps, Mapquest) and the terms of use for your map base layer must be compatible with our CC-BY 4.0 license. 

Additional Editor Comments (if provided):

Reviewers' comments:

Reviewer's Responses to Questions

**Comments to the Author**

1. Does this manuscript meet PLOS Global Public Health’s publication criteria ? Is the manuscript technically sound, and do the data support the conclusions? The manuscript must describe methodologically and ethically rigorous research with conclusions that are appropriately drawn based on the data presented.

Reviewer #1: Yes

Reviewer #2: Yes

2. Has the statistical analysis been performed appropriately and rigorously?

Reviewer #1: Yes

Reviewer #2: Yes

3. Have the authors made all data underlying the findings in their manuscript fully available (please refer to the Data Availability Statement at the start of the manuscript PDF file)?

Reviewer #1: Yes

Reviewer #2: Yes

4. Is the manuscript presented in an intelligible fashion and written in standard English?

Reviewer #1: Yes

Reviewer #2: Yes

5. Review Comments to the Author

Reviewer #1: Background:

Clarity and Focus: Streamline the background to emphasize the study's aims and its relevance to PCV10 and pneumonia outcomes among children under 5 in Uganda.

Scope and Context: Focus on the relationship between therapies like PCV10 and the study's objectives, particularly regarding pneumonia admissions and mortality.

Structure and Flow: Organize the background to logically flow from global concerns to the specific context of Uganda.

Methodology:

Data Quality: Provide more details on data quality and reliability from DHIS2 to bolster credibility.

Methodological Clarity: Offer clearer explanations of data extraction, verification, and analytical methods.

Geographical Coverage: Clearly distinguish between national and regional data to avoid confusion.

Data Analysis: Clarify the statistical methods used, including significance thresholds.

Ethical Considerations: Ensure the ethical considerations are thorough and highlight data protection measures.

Results:

Context: Offer context on possible confounding factors affecting trends.

Maps: Ensure visual data (e.g., maps) is clear and easy to interpret.

Statistical Analysis: Clarify statistical methods and their interpretation for better comprehension.

Discussion:

Clarification: Further clarify the impact of specific interventions on the observed trends.

Comparative Analysis: Directly compare findings with other African countries to highlight differences and similarities.

Regional Differences: Explore reasons for regional variations in pneumonia outcomes, providing hypotheses for further research.

Limitations: Present limitations succinctly and address potential biases or areas for improvement.

Further Recommendations: Suggest additional research avenues or data collection methods for deeper insight.

By incorporating these suggestions, the study can enhance its clarity, robustness, and overall contribution to understanding pneumonia outcomes in Uganda.

Reviewer #2: The study is robust, relevant and contributes to the body of scientific evidence. The authors studied the long term impact of PCV10 among under five children in Uganda using a national surveillance secondary data.

Comments

The title does not fully project the findings and significance of the study. Indeed a search for the purpose of systematic review regarding PCV impact could easily miss this paper when published.

I would suggest you re-craft the title to show at a glance that this paper is aimed at evaluating the long term impact impact of PCV10 among Ugandan under five children.

The authors mentioned ages for PCV vaccination as 6, 10 and 12 weeks; is this correct or did they mean 6, 10 and 14 weeks? Doses are given at least weeks apart or is Uganda giving third dose after 2 weeks?

The authors noted that the official rollout of PCV10 was in 2014, would it not be better to align analysis and title to 2014 instead of 2013 for uniformity?

Line 131 - "Although the PCV10 vaccine was introduced in 2013, official rollout to the entire country started in 201417. PCV 3 vaccine coverage was therefore calculated from 2014 to 2021." Everything should be calculated from 2014.

It is unclear if authors are assessing 2013-2021 or 2014-2021. When they say 9years, the period is not clear as the months in 2013/2014 and 2021 included in the study are unclear.

I suggest mentioning the months and limiting the entire analysis to 2014-2021, rather analysing coverage from 2014 and other analyses from 2013, and possibly reflecting this on the title.

The authors did not say anything about indirect effects. Would have been interesting to measure indirect effects on older age groups? This further amplifies the impact of PCV10 as has been documented elsewhere.

Discussion

Lines 205-207 - Generally good discussion on possible confounders especially of revised WHO admission guidelines, but many impact studies have shown positive results and Uganda may not be an exception especially with the high vaccination coverage and national data.

Line 226-235 - Were the cited studies on Vaccine strain replacement in u5 or adults?

There is need for further studies on serotypes and trends in older children and adults in Uganda.

Figures

Merge figs 1-3 as 1a-c

Merge figs 4-5 as fig 2a-b

6. PLOS authors have the option to publish the peer review history of their article (what does this mean? ). If published, this will include your full peer review and any attached files.

**Do you want your identity to be public for this peer review?** For information about this choice, including consent withdrawal, please see our Privacy Policy .

Reviewer #1: **Yes: ** Sarashwati Giri

Reviewer #2: **Yes: ** Beckie Nnenna Tagbo

---

## [Decision Letter · Decision Letter 1]

14 Nov 2024

Long-term impact of 10-valent pneumococcal conjugate vaccine among children <5 years, Uganda, 2014–2021.

PGPH-D-24-00245R1

Dear Dr. Wanyana,

We are pleased to inform you that your manuscript 'Long-term impact of 10-valent pneumococcal conjugate vaccine among children <5 years, Uganda, 2014–2021.' has been provisionally accepted for publication in PLOS Global Public Health.

Best regards,

Dandara de Oliveira Ramos, PhD

Academic Editor

Reviewer Comments (if any, and for reference):

Reviewer's Responses to Questions

**Comments to the Author**

1. If the authors have adequately addressed your comments raised in a previous round of review and you feel that this manuscript is now acceptable for publication, you may indicate that here to bypass the “Comments to the Author” section, enter your conflict of interest statement in the “Confidential to Editor” section, and submit your "Accept" recommendation.

Reviewer #1: All comments have been addressed

2. Does this manuscript meet PLOS Global Public Health’s publication criteria ? Is the manuscript technically sound, and do the data support the conclusions? The manuscript must describe methodologically and ethically rigorous research with conclusions that are appropriately drawn based on the data presented.

Reviewer #1: Yes

3. Has the statistical analysis been performed appropriately and rigorously?

Reviewer #1: Yes

4. Have the authors made all data underlying the findings in their manuscript fully available (please refer to the Data Availability Statement at the start of the manuscript PDF file)?

Reviewer #1: Yes

5. Is the manuscript presented in an intelligible fashion and written in standard English?

Reviewer #1: Yes

6. Review Comments to the Author

Reviewer #1: addressed all the comments

7. PLOS authors have the option to publish the peer review history of their article (what does this mean? ). If published, this will include your full peer review and any attached files.

**Do you want your identity to be public for this peer review?** For information about this choice, including consent withdrawal, please see our Privacy Policy .

Reviewer #1: **Yes: ** Sarashwati Giri
